# Curcumin-loaded nanocomplexes ameliorate the severity of nonalcoholic steatohepatitis in hamsters infected with *Opisthorchis viverrini*

**Chutima Sitthirach[1,2], Lakhanawan Charoensuk[2,3], Chawalit Pairojkul[2,4], Apisit Chaidee[1,2], Kitti Intuyod[2,4], Thatsanapong Pongking[2,5], Phonpilas Thongpon[1,2], Chanakan Jantawong[2,6], Nuttanan Hongsrichan[1,2], Sakda Waraasawapati[2,4], Manachai Yingklang[2,7], Somchai Pinlaor[1,2]***

1 Department of Parasitology, Faculty of Medicine, Khon Kaen University, Khon Kaen, Thailand, 2 Cholangiocarcinoma Research Institute, Khon Kaen University, Khon Kaen, Thailand, 3 Department of Clinical Pathology, Faculty of Medicine Vajira Hospital, Navamindradhiraj University, Bangkok, Thailand, 4 Department of Pathology, Faculty of Medicine, Khon Kaen University, Khon Kaen, Thailand, 5 Biomedical Science Program, Graduate School, Khon Kaen University, Khon Kaen, Thailand, 6 Department of Medical Technology, Faculty of Allied Health Science, Nakhonratchasima College, Nakhon Ratchasima, Thailand, 7 Department of Fundamentals of Public Health, Faculty of Public Health, Burapha University, Chonburi, Thailand

* psomec@kku.ac.th

## Abstract

### Background

Comorbidity of *Opisthorchis viverrini* (OV) infection and nonalcoholic fatty-liver disease (NAFLD) enhances NAFLD progression to nonalcoholic steatohepatitis (NASH) by promoting severe liver inflammation and fibrosis. Here, we investigated the effect of supplementation with curcumin-loaded nanocomplexes (CNCs) on the severity of NASH in hamsters.

### Methodology

Hamsters were placed in experimental groups as follows: fed standard chow diet (normal control, NC); fed only high-fat and high-fructose (HFF) diet; *O. viverrini*-infected and fed HFF diet (HFFOV); group fed with blank nanocomplexes (HFFOV+BNCs); groups fed different doses of CNCs (25, 50 and 100 mg/kg body weight: HFFOV+CNCs25; HFFOV+CNCs50; HFFOV+CNCs100, respectively) and a group given native curcumin (HFFOV+CUR). All treatment were for three months.

### Results

The HFF group revealed NAFLD as evidenced by hepatic fat accumulation, ballooning, mild inflammation and little or no fibrosis. These changes were more obvious in the HFFOV group, indicating development of NASH. In contrast, in the HFFOV+CNCs50 group, histopathological features indicated that hepatic fat accumulation, cell ballooning, cell inflammation and fibrosis were lower than in other treatment groups. Relevantly, the expression of lipid-uptake genes, including fatty-acid uptake (cluster of differentiation 36), was reduced, which was associated with the lowering of alanine aminotransferase, total cholesterol and

**Data Availability Statement:** All relevant data are within the manuscript and its Supporting information files.

**Funding:** This study was supported by the Research and Researchers for Industries (RRI: MSD62I0041), Thailand Science Research and Innovation (TSRI: RDG6250045) and Mekong Health Science Research Institute Khon Kaen University (MeHSRI09/2561). The funders had no role in study design, data collection and analysis, decision to publish, or preparation of the manuscript.

**Competing interests:** The authors have declared that no competing interests exist.

triglyceride (TG) levels. Reduced expression of an inflammation marker (high-mobility group box protein 1) and a fibrosis marker (alpha smooth-muscle actin) were also observed in the HFFOV+CNCs50 group.

## Conclusion

CNCs treatment attenuates the severity of NASH by decreasing hepatic steatosis, inflammation, and fibrosis as well as TG synthesis. CNCs mitigate the severity of NASH in this preclinical study, which indicates promise for future use in patients.

## Introduction

Nonalcoholic fatty liver disease (NAFLD) is caused by excess triglyceride accumulation in the liver cells in the absence of alcohol consumption [1]. NAFLD is the most common chronic liver disease worldwide, occurring in about 25% of the global population [2]. The interactions of environmental and genetic factors are associated with risk for NAFLD development [3, 4]. NAFLD frequently progresses to nonalcoholic steatohepatitis (NASH), cirrhosis and hepatocellular carcinoma or cholangiocarcinoma (CCA). Liver fat accumulation, hepatic inflammation and fibrosis all play a vital role in this [5, 6]. Chemical promotion of NAFLD progression to NASH is well known [7, 8]; however, the role of infectious agents in driving disease progression is unclear.

In northeast Thailand, the incidence of fatty-liver disease is on the rise and infection with the small liver fluke, *Opisthorchis viverrini*, is common [9]. Infection with *O. viverrini* is one of the risk factors for periductal fibrosis and chronic inflammation, leading to hepatobiliary diseases including CCA [10]. Prior experimental study has demonstrated that *O. viverrini* infection and high-fat diets can rapidly accelerate the progression from NAFLD to NASH [11]. Although praziquantel is an effective drug for treating *O. viverrini* infection, re-infection following treatment is common in endemic areas, leading to persistence of a high prevalence of infection [12] and severe hepatobiliary diseases, including fatty liver [13]. In addition, delayed treatment [14], side effects following early treatment [15] and frequent treatments [16] are all likely to enhance risk of severe hepatobiliary diseases. There are no specific and effective drugs for NAFLD treatment [17]. Phytomedicine or herbal substances, which have few side effects and low toxicities, may offer alternative treatments to prevent NAFLD progression [18].

Curcumin is the principal active constituent in the rhizome of *Curcuma longa* L. (turmeric). It has been extensively studied because of its pharmacological properties, especially its antioxidant, anti-fibrosis, anti-inflammatory and anti-lipidemia activities [19, 20]. The efficacy of curcumin against bile-duct cancer, hepatobiliary disease and NAFLD has also been described [21, 22]. Use of curcumin is typically limited by its water insolubility, instability and poor bioavailability [23]; however, nanocarriers, especially involving encapsulation, have been developed to solve these limitations [24].

Treatment with nano-curcumin leads to improved glucose indices, lipids, inflammation, and nesfatin in overweight and obese patients with NAFLD [25]. We also developed a novel nano-encapsulated curcumin based on a polymeric nanocarrier (called curcumin-loaded nanocomplexes, CNCs) that is safe and has no obvious side effects during either short-term or long-term consumption in animal models [26, 27]. However, the hepatoprotective effects of CNCs on NASH progression have not been evaluated.

In this study, we aimed to investigate the effects of CNCs on the severity of NASH induced by high-fat and high-fructose diets and *O. viverrini* infection in hamsters. Histopathological

study, biochemical assays, western blotting and real-time PCR analysis were used. This pre-clinical study on the ability of CNCs to alleviate the progression of NASH might be translatable to future clinical use.

# Materials and methods

## Ethic statement

This experimental protocol was reviewed and approved by the Animal Ethic Committee of Khon Kaen University (IACUC–KKU–81/62). Hamsters were obtained from the Animal Unit, Faculty of Medicine, Khon Kaen University.

## Experimental design

Eighty male Syrian golden hamsters (*Mesocricetus auratus*: 4–6 weeks old and 80–100 g) were used for the experiment. Hamsters were divided into 8 groups (n = 10 each) as shown in Table 1.

The high-fat diet and high-fructose drinking water were prepared freshly every week and hamsters were fed on this diet all the time. In relevant treatment groups, the BNCs, CNCs and native curcumin were orally administered to hamsters using a gastric tube thrice weekly and continued until the animals were sacrificed at the end of 3 months. Hamsters were anesthetized with isoflurane and sacrificed by drawing blood directly from the heart. Afterwards, liver and blood-serum samples were collected. Sera were stored at -20˚C until used to determine biochemical parameters. A portion of each liver sample was preserved in 10% formalin for histopathological and immunohistochemical analysis. The remainder of each liver sample was snap frozen in liquid nitrogen (for western-blot analysis) and immediately treated with TRIzol reagent (Invitrogen, Carlsbad, CA, USA) for total RNA isolation. Frozen livers and preserved liver in TRIzol reagent samples were kept at -80˚C until use.

## *Opisthorchis viverrini* infection

*Opisthorchis viverrini* metacercaria were obtained from naturally infected cyprinid freshwater fish. The protocol, reagents and procedure for fish digestion were as described previously [28]. Fifty live *O. viverrini* metacercariae were administered to each hamster by oral inoculation using gastric intubation [11].

**Table 1. Experimental design and animal groups.**

| Group no. | Treatment | Group abbreviation |
|---|---|---|
| 1 | Normal controls: not infected with *O. viverrini* (OV) and fed standard chow diet (Smart Heart, PCG, Bangkok, Thailand) and filtered water. | NC |
| 2 | Fed high-fat, high-fructose (HFF) diet, no OV infection. | HFF |
| 3 | Fed HFF diet and infected with OV. | HFFOV |
| 4 | Fed HFF diet, infected with OV, fed blank nanocomplexes. | HFFOV+BNCs |
| 5 | Fed HFF diet, infected with OV, fed curcumin-loaded nanocomplexes at 25mg/kg bw (equivalent to curcumin 6.25 mg/kg bw). | HFFOV+CNCs25 |
| 6 | Fed HFF diet, infected with OV, fed curcumin-loaded nanocomplexes at 50 mg/kg bw (equivalent to curcumin 12.5 mg/kg bw). | HFFOV+CNCs50 |
| 7 | Fed HFF diet, infected with OV, fed curcumin-loaded nanocomplexes at 100 mg/kg bw (equivalent to curcumin 25 mg/kg bw). | HFFOV+CNCs100 |
| 8 | Fed HFF diet, infected with OV, fed native curcumin at 25 mg/kg bw. | HFFOV+CUR |

## Preparation of curcumin, CNCs and BNCs

The curcumin powder (97% purity w/w) was purchased from Merck-Schuchardt (Hohenbrunn, Germany). Powdered CNCs (containing 28.6% curcumin by weight) and blank nanocomplexes (BNCs) were obtained as in the previous study [26, 27]. Curcumin powder was weighed and dissolved in corn oil, and BNCs and CNCs powders were dispersed in distilled water [29]. The final concentrations of CNCs used were 25, 50, and 100 mg/kg bw, which contained native curcumin at 6.5, 12.5 and 25 mg/kg bw, respectively. BNCs were administered to the appropriate groups at 100 mg/kg bw.

## Preparation of the high-fat diet and high-fructose drinking water

The high-fat diet used was modified from that described previously [11]. The ingredients were a mixture of 40% control diet (Smart Heart; PCG, Bangkok, Thailand), 10% coconut oil (Roi Thai, Thailand), 10% corn oil (Golden Drop, Thailand), 1.25% cholesterol, 0.25% sodium deoxycholate (Sigma-Aldrich, St, Louis, Mo, USA), and 38.5% sucrose (Mitr Phol, Thailand). All of the ingredients were combined and formed into small discs, which were then incubated at 65˚C for 24 h. The high-fructose drinking water was prepared by mixing d-glucose powder (18.9 g) and d-fructose powder (23.1 g) (Merck Millipore, Darmstadt, Germany) in distilled water (1 L) [30]. HFF diet composition was determined by Central Laboratory (Thailand), Co. Ltd. according to the Association of Office Analytical Collaboration (AOAC) International guidelines and is shown in S1 Table.

## Measurement of serum biochemistry parameters

Biochemical parameters of liver function (alanine aminotransferase: ALT) and lipid profiles (total cholesterol; TC; and triglyceride: TG levels) were measured at the end of the experiment from serum using automated analyzers (Cobas 8000 Modular Analysis Series, Roche Diagnostics International Ltd) at the clinical laboratory of Srinagarind Hospital, Faculty of Medicine, Khon Kaen University, Thailand.

## Investigation of histopathological changes to the liver

To investigate the histopathology of liver changes (fat accumulation, ballooning, inflammation, and fibrosis), liver sections were stained with Mayer's hematoxylin and eosin (H&E) and the picrosirius-red method was applied. The protocol has been described elsewhere [29]. In brief, formalin-fixed liver samples were embedded in paraffin and cut into 5 μm thick sections using a rotary microtome HM 315 (Thermo Fisher Scientific, USA). The sections were deparaffinized in xylene and rehydrated in a descending ethanol series. Afterwards, slides were stained with Mayer's hematoxylin and eosin for 10 and for 5 min, respectively. To observe fibrosis, the picrosirius red staining method was used (Solution A for 2 min, Solution B for 90 min, and Solution C for 2 min) and stained with Mayer's hematoxylin 8 min. Finally, slides were washed with distilled water until clear-colored and dehydrated in an ascending ethanol series followed by xylene.

Histopathological grading of NAFLD or NASH indicators (fat accumulation, ballooning, inflammation and fibrosis) was scored on a scale of 0–3, following previous studies [11, 31, 32]. Stained slides (7–8 hamsters per each group) were randomly selected and the lobular area at the periphery and perihilar regions of the liver were examined. The histopathological grading was performed by three independent researchers (agreement of two out of three being accepted), and the grading scores were confirmed by a senior clinical pathologist under light microscopy using 5X to 40X magnifications. The criteria for staging are shown in Table 2. All

**Table 2. The histological scoring system of hepatic lobules in NAFLD modified from [11, 31, 32].**

| Grade | Lobular area at periphery and perihilar region of the liver |
|---|---|
| **Fatty change grade** | |
| 0 | None |
| 1 | Dominant micro-vesicular steatosis |
| 2 | Mixed micro- and macro-vesicular steatosis |
| 3 | Dominant macro-vesicular steatosis |
| **Ballooning grade** | |
| 0 | None |
| 1 | Pericentral regions |
| 2 | Periportal regions |
| 3 | Mixed (pericentral and periportal) regions |
| **Inflammation grade** | |
| 0 | None |
| 1 | Pericentral inflammation |
| 2 | Periportal inflammation |
| 3 | Mixed pericentral and periportal inflammation |
| **Fibrosis stage** | |
| 0 | None |
| 1 | Perisinusoidal/pericellular fibrosis |
| 2 | Periportal/portal fibrosis |
| 3 | Bridging fibrosis |

sections were examined for the characteristics of NAFLD/NASH to allow visualization of grading indicators (fat accumulation, ballooning, inflammation and fibrosis (see Fig 1). The scored grades were then applied to the algorithm for NAFLD/NASH diagnosis (shown in Fig 2).

## Immunohistochemistry assays

Tissue sections cut at 5 μm thickness were deparaffinized in xylene and rehydrated in a descending ethanol series and then the sections were autoclaved at 110˚C for 10 min in citrate buffer (10 mM sodium citrate, 0.05% Tween 20, pH 6.0) to retrieve antigen. Thereafter, slides were immersed in 3% $H_2O_2$ for 10 min to quench endogenous peroxidases. Non-specific binding was blocked by 5% bovine serum albumin (BSA) for 1 h at room temperature. The slides were then incubated with primary antibodies, rabbit polyclonal anti-high-mobility group box protein 1 or HMGB–1 (1:300, ab79823, Abcam, Cambridge, MA, USA), and mouse monoclonal anti-alpha smooth-muscle actin antibody or α–SMA (1:100, ab7817, Abcam) at 4˚C overnight. Afterwards, slides were washed with phosphate-buffered saline solution (PBS) for 5 min, followed by incubation with a 1:200 dilution of horseradish peroxidase (HRP)-labeled goat anti-rabbit IgG (Jackson ImmunoResearch, West Grove, PA, USA) and a 1:200 dilution of HRP-conjugated sheep anti-mouse IgG (Jackson ImmunoResearch) at room temperature for 1 h. Then, the immunoreactivity was visualized by adding DAB solution (3', 3'-diaminobenzidine) (0.02%) in 0.05 M Tris–HCl (pH 7.6) and 0.01% $H_2O_2$ (v/v) was used as a chromogenic substrate for 20 s. Slides were counterstained with Mayer's hematoxylin for 2 min, and then washed with distilled water and dehydrated with ethanol (70, 95 and 100%) and xylene. The slides were examined under a light microscope using 20X magnification (Carl Zeiss, Jena, Germany). Ten representative, randomly selected areas of the liver, were monitored. Fibrosis and inflammation-cell scores were analyzed using ImageJ software (National Institutes of Health, Bethesda, MD, USA).

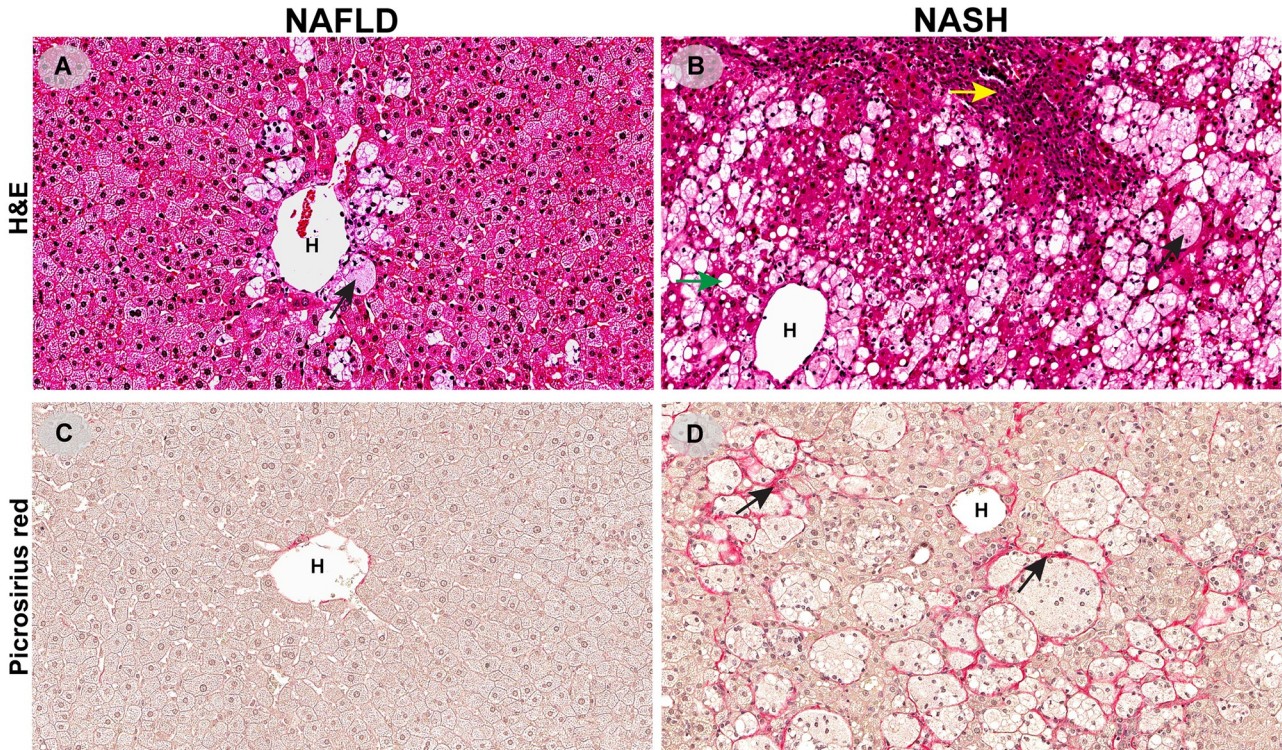

**Fig 1. Characteristics of non-alcoholic fatty liver disease (NAFLD, A and C) and non-alcoholic steatohepatitis (NASH, B and D).** Liver tissues were stained with hematoxylin and eosin (H&E, A and B) and picrosirius red (C and D). A and C illustrate NAFLD. A): In this case of mild NAFLD steatosis, scattered micro-vesicles are seen in the hepatic lobules and with a predominantly perivenular distribution associated with ballooning (black arrow). C): Fibrosis is almost absent in NAFLD between lobules. B and D represent NASH. B): Macro-vesicular steatosis is scattered in the hepatic lobule (green arrow) and swollen (ballooned) hepatocytes are apparent with rarefied cytoplasm (black arrow). Periportal inflammation can be seen as a mixed inflammatory infiltrate consisted of mainly mononuclear cells as shown (yellow arrow). D): Pericellular and perisinusoidal lobular fibrosis (black arrows). H, hepatic vein.

## Western blot analysis

Protein was extracted from frozen liver with lysis buffer until homogenous, and then centrifuged at 13000 rpm at 4°C for 10 min. The supernatant was collected and protein concentration measured using the BCA assay with an ELISA micro-plate reader. Approximately 20 µg of liver protein was separated by SDS–PAGE and the resolved proteins were transferred onto a polyvinylidene difluoride membrane (PVDF). The membranes were blocked and incubated with primary antibody (α–SMA, 1:1000, ab7817, Abcam): diluted in 5% BSA) at 4°C overnight using gentle shaking. Afterwards, the membrane was washed with Tris-buffered saline with 0.1% Tween 20 detergent buffer and incubated with a secondary antibody 1:3000 dilution of HRP-conjugated sheep anti-mouse IgG (Jackson ImmunoResearch) for 1 h. An enhanced chemiluminescence detection reagent (ECL Prime, GE Healthcare) was added onto the membrane and then the developed immunoreactive band was visualized using an Image Quant LAS4000 mini machine (GE Healthcare Bio-Sciences AB, Uppsala, Sweden). Relative band intensity was measured using ImageJ software (National Institutes of Health, Bethesda, MD, USA).

## RNA extraction and quantitative real-time PCR

Total RNA was extracted from frozen liver samples with TRIzol reagent (Invitrogen, Carlsbad, CA, USA). The quality of RNA was checked using a Nanodrop 2000 (NanoDrop Technologies,

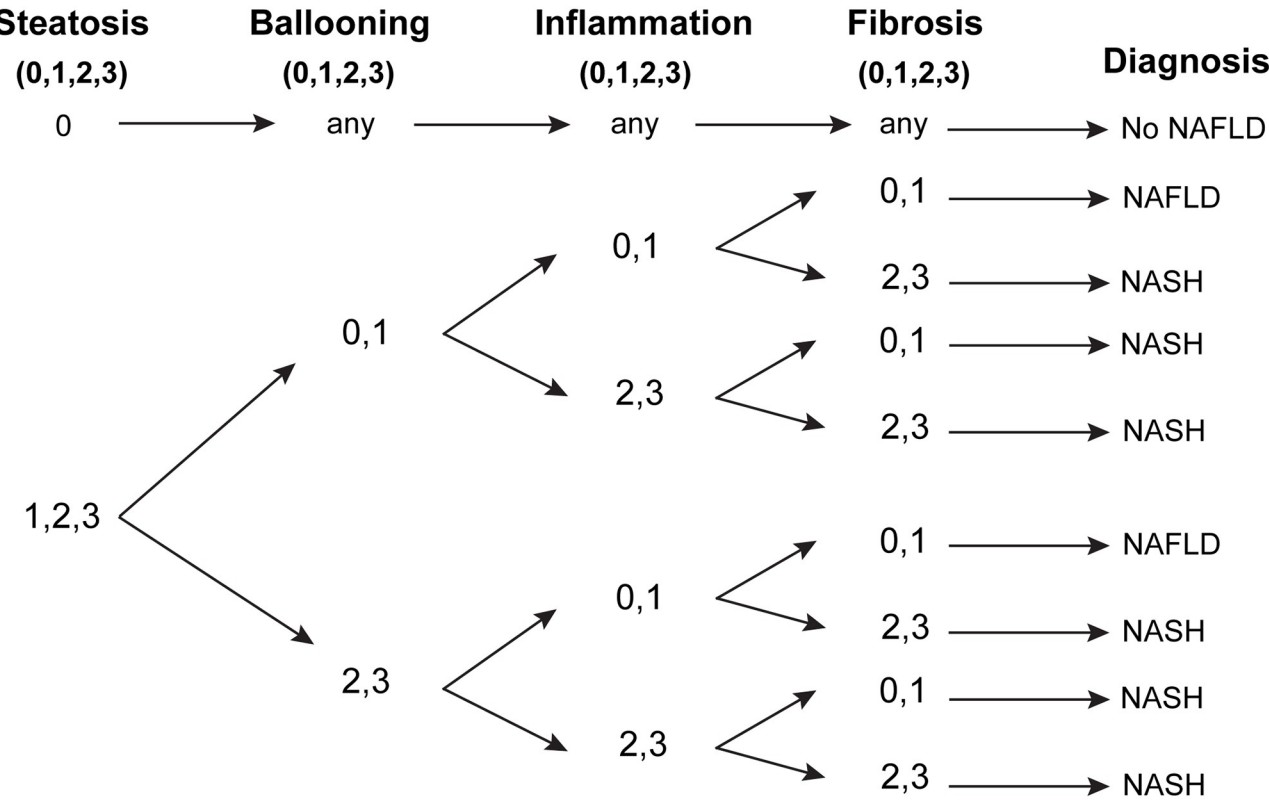

**Fig 2. Diagnostic algorithm for differentiation of NAFLD and NASH at the liver parenchyma [31].**

Wilmington, DE, USA). RNA was reverse transcribed into cDNA using the Revert Aid First Strand cDNA Synthesis Kit (K1621, Molecular Biology, Thermo Scientific), following the manufacturer's protocol.

To detect free fatty-acid uptake and lipogenesis, the genes for cluster differentiation 36 (CD36), sterol regulatory element-binding protein 1c (SREBP–1c), and fatty-acid synthase (FAS) were assessed. The specific primer pairs for the CD36 and FAS genes were designed based on sequence information for complete CD36 mRNA (accession number: U42430.1) and FAS mRNA (accession number: XM_005069786.4) of *Mesocricetus auratus* in the GenBank database using online primer 3 software (https://bioinfo.ut.ee/primer3-0.4.0/). The primer pair used for SREBP–1c was as previously described [33]. The specific primer pair for the free fatty-acid uptake gene (CD36) was as follows: F; 5´ ACGACTACATTTACGCACTGG 3´, R; 5´ TTGAAATATGCTTTGGCTTAGTGC 3´. The primer pairs for lipogenesis detection were as follows: F: 5´ CAGCTCAGAGCCGTGGTGA 3´ R; 5´ TTGATAGAAGACCGGTAGCGC 3´ for the SREBP-1c gene and F; 5´ CCATCATCCCCTTGATGAAGA 3´ and R; 5´ GTTGATGT CGATGCCTGTGAG 3´ for the FAS gene. The RT-PCR reaction (total volume 15 μL) included FastStart Universal SYBR Green Master mix (ROX, Roche Applied Science, Mannheim, Germany) and 5 μg of cDNA template. Cycling conditions in a Light Cycler 480 II system (Roche Applied Science, Mannheim, Germany) were as follows: annealing at 95˚C for 10 min; 40 cycles of 95˚C for 15 s, 60˚C and 60˚C and 61˚C for the CD36, SREBP–1c, and FAS genes (respectively) for 30 s, and extension at 72˚C for 1 min. Relative mRNA expression was calibrated using the $2^{\Delta\Delta CT}$ method, and the beta-actin (β-actin) gene was used as a calibrator. The determination of mRNA expression was conducted in duplicate.

## Statistical analysis

Data from the eight groups in the experiment were presented as the mean ± standard deviation, and differences between groups were compared using one-way analysis of covariance (ANOVA) with the Tukey HSD post-hoc test. A nonparametric Mann-Whitney U test was used to compare the graded score and nonparametric data. These analyses were performed using IBM SPSS statistics version 26 (Armonk, NY: IBM Corp).

## Results

### Effect of CNCs on body weight and morphology of liver changes

The average body weight of hamsters in each treatment group at three months did not statistically significantly differ from that of the un-treated group (HFFOV). The average liver weight in the HFFOV group was significantly higher than in the HFF group ($P< 0.05$). In contrast, the liver weights in the HFFOV+CUR, HFFOV+BNCs and the three curcumin-loaded nanocomplex (CNCs) treatment groups (25, 50, and 100 mg/kg bw) did not significantly differ from weights in the HFFOV group (Table 3). The livers from all high-fat, high-fructose groups, whether supplemented or not with curcumin, BNCs, or CNCs (at 25, 50, and 100 mg/kg bw), were markedly enlarged and pale in color when compared to the NC group (Fig 3).

### Effect of CNCs on biochemical parameters

At the end of three months post-treatment, as shown in Table 3, the HFFOV group displayed significantly higher levels of ALT, TC and TG than did the HFF group ($P< 0.05$). In contrast, the level of ALT was significantly decreased in the treatment groups with CNCs at 25, and 50 mg/kg bw and BNCs, while treatment with CNCs at 100 mg/kg bw and native curcumin did not show a significant difference when compared to the HFFOV group (un-treated group). The level of TC was also significantly lower than in the HFFOV group after treatment with all regimens, while the level of TG was significantly lower only in the treatment group with CNCs (at 25, 50, and 100 mg/kg bw) and native curcumin ($P< 0.05$).

**Table 3. Parameters relating to NAFLD/NASH disease of hamsters in the experimental groups at the end of three months.**

| Parameters | NC | HFF | HFFOV | HFFOV+ BNCs | HFFOV+ CNCs25 | HFFOV+ CNCs50 | HFFOV+ CNCs100 | HFFOV+ CUR |
|---|---|---|---|---|---|---|---|---|
| Body weight (g) | 158±15.38 | 136±22.34 | 172±23.45 | 182±10.17 | 166±13.24 | 171±23.22 | 173±30.41 | 177±25.76 |
| Liver weight (g) | 4±0.37 | 8±1.12 [a] | 10±1.96 [b] | 10±0.94 | 9±0.59 | 10±1.70 | 10±2.24 | 9±1.66 |
| LW/BW (Ratio) | 0.02±0.002 | 0.06±0.010 [a] | 0.06±0.060 | 0.05±0.005 | 0.05±0.006 | 0.06±0.014 | 0.06±0.017 | 0.05±0.007 |
| ALT(U/L) | 46±7.18 | 69±13.89 | 115±21.40 [b] | 87±8.72 [c] | 85±28.71 [c] | 80±8.62 [c] | 89±14.23 | 95±23.97 |
| TC (mg/dL) | 69±11.69 | 189±35.81 [a] | 279±30.97 [b] | 205±4.32 [c] | 168±3.80 [c] | 192±6.51 [c] | 241±4.72 [c] | 177±26.84 [c, g] |
| TG (mg/dL) | 93±4.03 | 121±23.30 | 148±39.56 | 136±22.92 | 96±13.78 [c] | 110±18.29 [c] | 88±15.61 [c] | 96±17.98 [c] |

Liver weight/body weight: LW/BW; Alanine aminotransferase: ALT, Total cholesterol: TC; Triglyceride: TG.

The differences between groups were compared using one-way analysis of covariance (ANOVA) with the Tukey HSD post-hoc test. Any *P*-value of less than 0.05 was considered to indicate statistical significance.

[a] *P*<0.05 when compared with NC,

[b] *P*<0.05 when compared with HFF,

[c] *P*<0.05 when compared with HFFOV,

[d] *P*<0.05 when compared with HFFOV+BNCs,

[e] *P*<0.05 when compared with HFFOV+CNCs25,

[f] *P*<0.05 when compared with HFFOV+CNCs50,

[g] *P*<0.05 when compared with HFFOV+CNCs100.

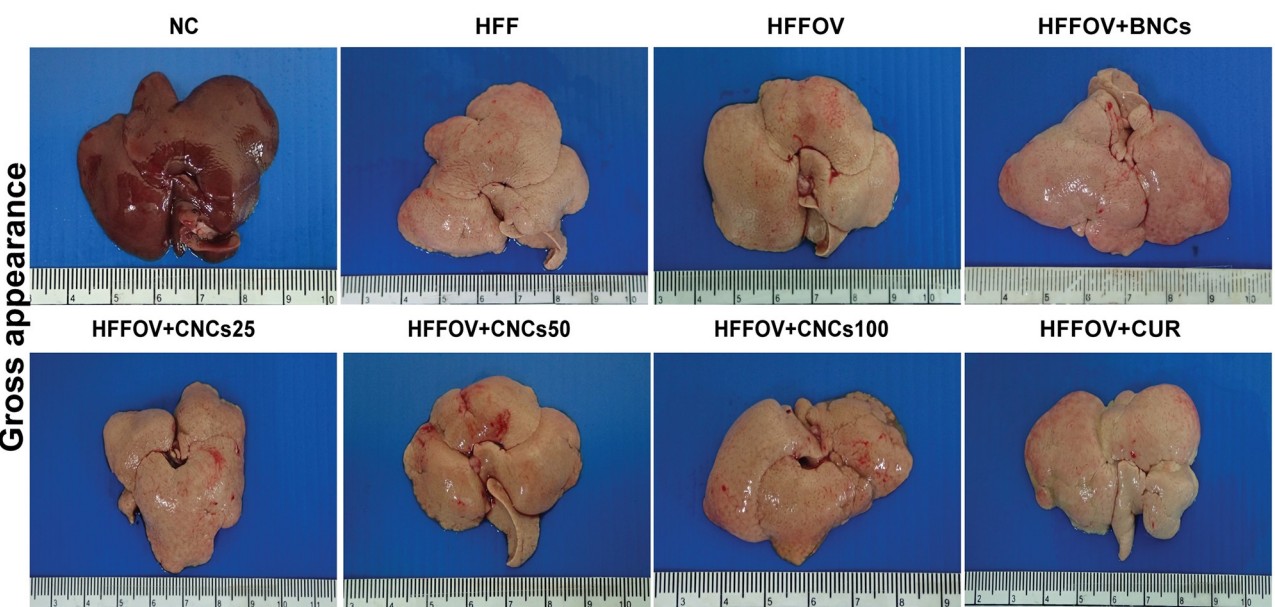

**Fig 3. Morphological gross appearance of hamster livers.** Normal hamsters (NC); hamsters treated with high-fat/high-fructose diet only (HFF); *O. viverrini*-infected hamsters and fed the HFF diet (HFFOV); *O. viverrini*-infected with HFF diet and supplemented with blank nanocomplexes (HFFOV +BNCs); *O. viverrini*-infected hamsters with HFF diet and supplemented with curcumin-loading nanocomplexes (CNCs) 25, 50, 100 mg/kg bw; and *O. viverrini*-infected hamsters with HFF diet; and supplemented with curcumin (HFFOV+CUR).

## Effect of CNCs on histopathological changes of liver

The histopathological changes in the hepatic lobules of liver are shown in Figs 4 and 5. Using H&E and picrosirius-red staining (Fig 4A and 4B), the level of fat-droplet accumulation (micro vesicular), hepatocyte ballooning, slight to mild inflammatory-cell aggregation, and fibrosis in hepatic lobules of the liver were observed in the HFF group, indicating that NAFLD had been successfully induced in this model. These histological changes, particularly macro-vesicular changes, inflammation and liver fibrosis were more pronounced in the HFFOV group than in the HFF group (Table 4 and Fig 4), indicating NASH. After three months of treatment, all groups receiving supplements (CNCs, BNCs or CUR) showed a slight tendency to decreased fat accumulation, ballooning, inflammation, pericellular fibrosis, the exception being periportal fibrosis in hepatic lobules (Table 4 and Fig 5). Although the CNCs treatment groups exhibited decreased severity of hepatitis, as shown by reduction of fat accumulation and inflammatory-cell infiltration, fibrosis was still present: therefore, this condition was still classified as NASH.

## Effect of CNCs treatment on the inflammation and fibrosis indicators in liver

Fig 6 depicts the immunochemistry staining (HMGB–1 and α–SMA expression) and the relative intensity of liver-protein expression using western-blot analysis (α–SMA expression). The expression levels of HMGB–1 and α–SMA were significantly higher in the HFFOV group than in the HFF group (Fig 6A and 6B). In contrast, the level of HMGB–1 was significantly lower in the group treated with CNCs (at 25, 50, and 100 mg/kg bw), while treatments with BNCs and curcumin did not lead to significant differences when compared to the un-treated group (HFFOV). The level of α–SMA was significantly decreased after treatment with CNCs (at 25 and 50 mg/kg bw), BNCs and curcumin, while treatment with CNCs (at 100 mg/kg bw) did

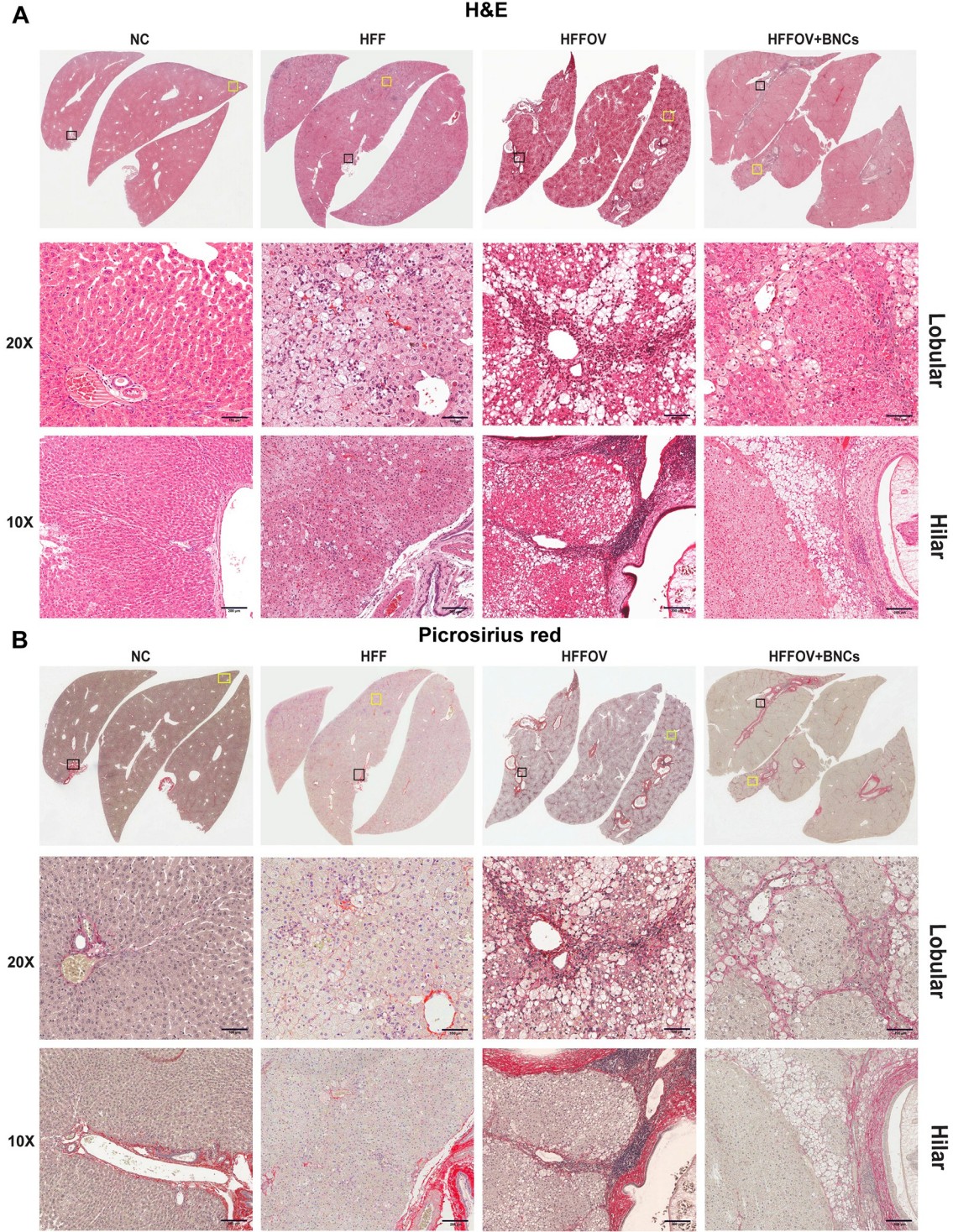

**Fig 4. Histopathological changes induced by *O. viverrini* infection and HFF diet.** A): H&E stain was used to demonstrate abnormalities in the liver; and B): Picrosirius red stain was used to investigate fibrosis. Experimental animal groups and abbreviations are the same as in Fig 3 legend.

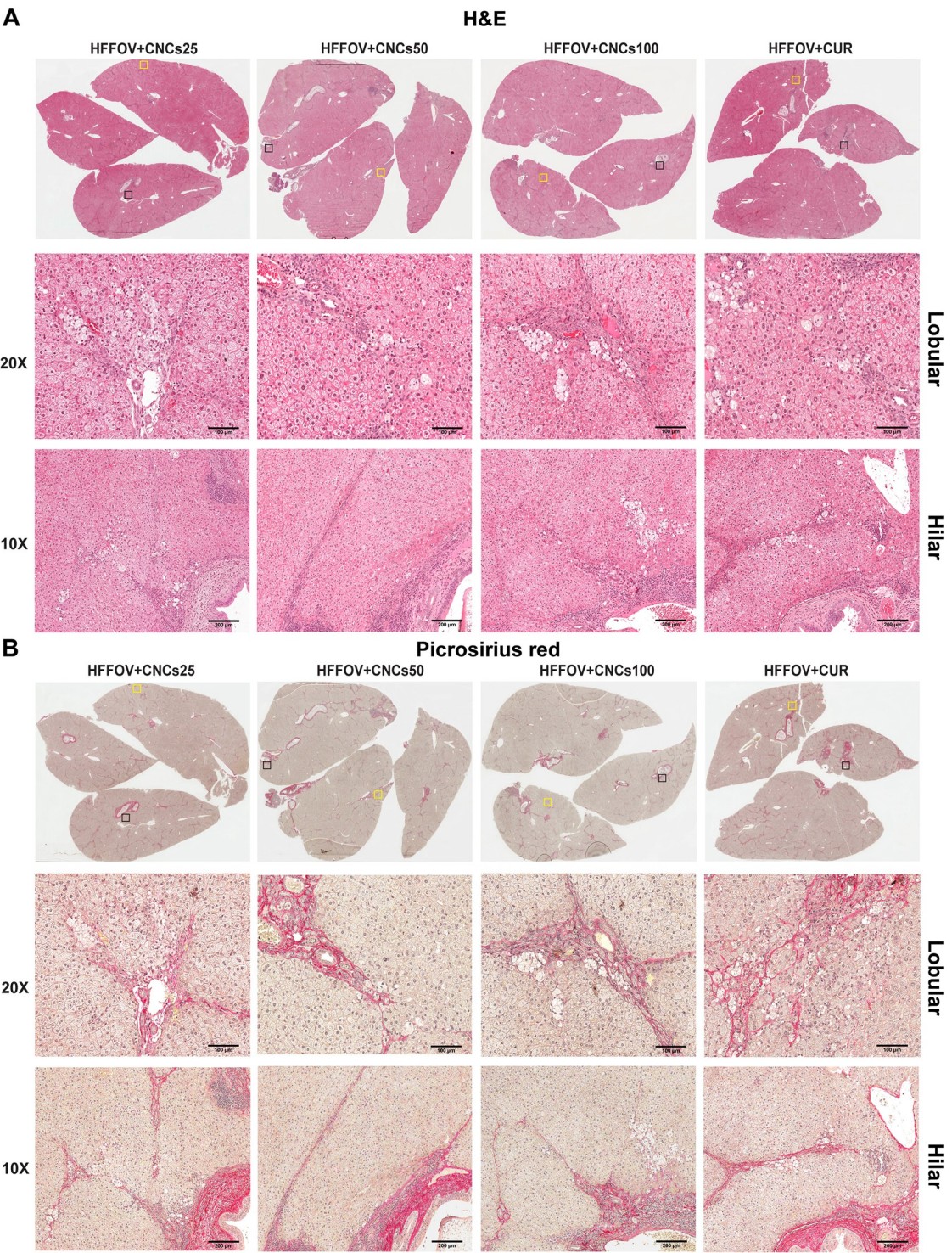

**Fig 5. Histopathological changes induced by *O. viverrini* infection and HFF diet in experimental groups that were also treated with CNCs.** A): H&E stain was used to investigate liver histopathological changes; and B): Picrosirius red stain was used to investigate fibrosis. Experimental animal groups and abbreviations are the same as in Fig 3 legend.

**Table 4. Comparisons of liver pathology grades (ranging from 0 to 3 in each case) among the experimental groups.**

| Characteristics | NC | HFF | HFFOV | HFFOV+ BNCs | HFFOV+ CNCs25 | HFFOV+ CNCs50 | HFFOV+ CNCs100 | HFFOV+ CUR |
|---|---|---|---|---|---|---|---|---|
| Steatosis grade | 0 | 1 | 2–3 | 1 | 1 | 1 | 1 | 1–2 |
| Ballooning grade | 0 | 1 | 3 | 1–2 | 1–2 | 1–2 | 1–2 | 1–2 |
| Inflammation grade | 0 | 1 | 2–3 | 2 | 2 | 1–2 | 2 | 2–3 |
| Fibrosis stage | 0 | 1 | 3 | 3 | 2–3 | 2–3 | 3 | 3 |
| Diagnosis | No NAFLD | NAFLD | NASH | NASH | NASH | NASH | NASH | NASH |

*Lobular and hilar periphery differences are not significant.

not show significant differences when compared to the HFFOV group (*P*> 0.05). According to the western-blot analysis, the groups treated with CNCs at 25 and 50 mg/kg bw, BNCs and curcumin exhibited a lower relative intensity of α–SMA when compared to the HFFOV group (Fig 6C), but this was not statistically significant.

## Effect of CNCs treatment on expression of genes associated with free fatty-acid uptake and lipogenesis

The relative expression levels of CD36, SREBP–1c, and FAS genes, determined using RT-PCR, are shown in Fig 7. The expression level of CD36 in the HFFOV group was significantly higher than in the HFF group. On the other hand, it was significantly lower in the CNCs-treated groups (all dose rates) and CUR group when compared with the HFFOV group (Fig 7A). The SREBP-1c gene was more highly expressed in all treatment groups compared to the un-treated group (HFFOV), with the highest expression in the group treated with CNCs at 50 mg/kg bw, followed by those treated with CNCs at 100 and with 25 mg/kg bw, native curcumin and BNCs, in that order (Fig 7B). The expression level of the FAS gene was highest in the groups treated with CNCs at 100 and 50 mg/kg bw (Fig 7C).

## Discussion

In this study, NAFLD and NASH were successfully induced in hamsters after three months by administration of a high-fat and high-fructose diet and infection with *O. viverrini*. This agrees with a previous study by Chaidee et al., 2019 [11]. These conditions were apparent in the HFFOV group, which exhibited severe liver pathology including macro-vesicular/micro-vesicular steatosis, hepatocellular ballooning, periductal inflammation, and fibrosis relative to the HFF group. However, hamsters treated with CNCs, relative to the HFFOV group, exhibited reduced hepatic steatosis and decreased expression of genes involved in free fatty-acid uptake, inflammation and fibrosis. Alleviation of histopathological features were consistent with the lower serum levels of ALT, TC, and TG parameters. Treatment with CNCs mitigates the severity of NASH by reducing expression of genes associated with the uptake of free fatty acids (CD36), liver inflammation (HMGB–1) and fibrosis (α–SMA). The potential mechanism by which CNCs ameliorate NASH in hamsters infected with *Opisthorchis viverrini* is shown in Fig 8.

By the end of the three-month experiment, there was no statistically significant difference in the average body weight or the liver weight of hamsters from all treatment groups when compared to the HFFOV group. The livers from all groups fed with the high-fat, high-fructose diet, whether treated with native curcumin, BNCs, or CNCs (at 25, 50, and 100 mg/kg bw) or un-treated (HFF and HFFOV), were enlarged relative to normal controls and pale in color.

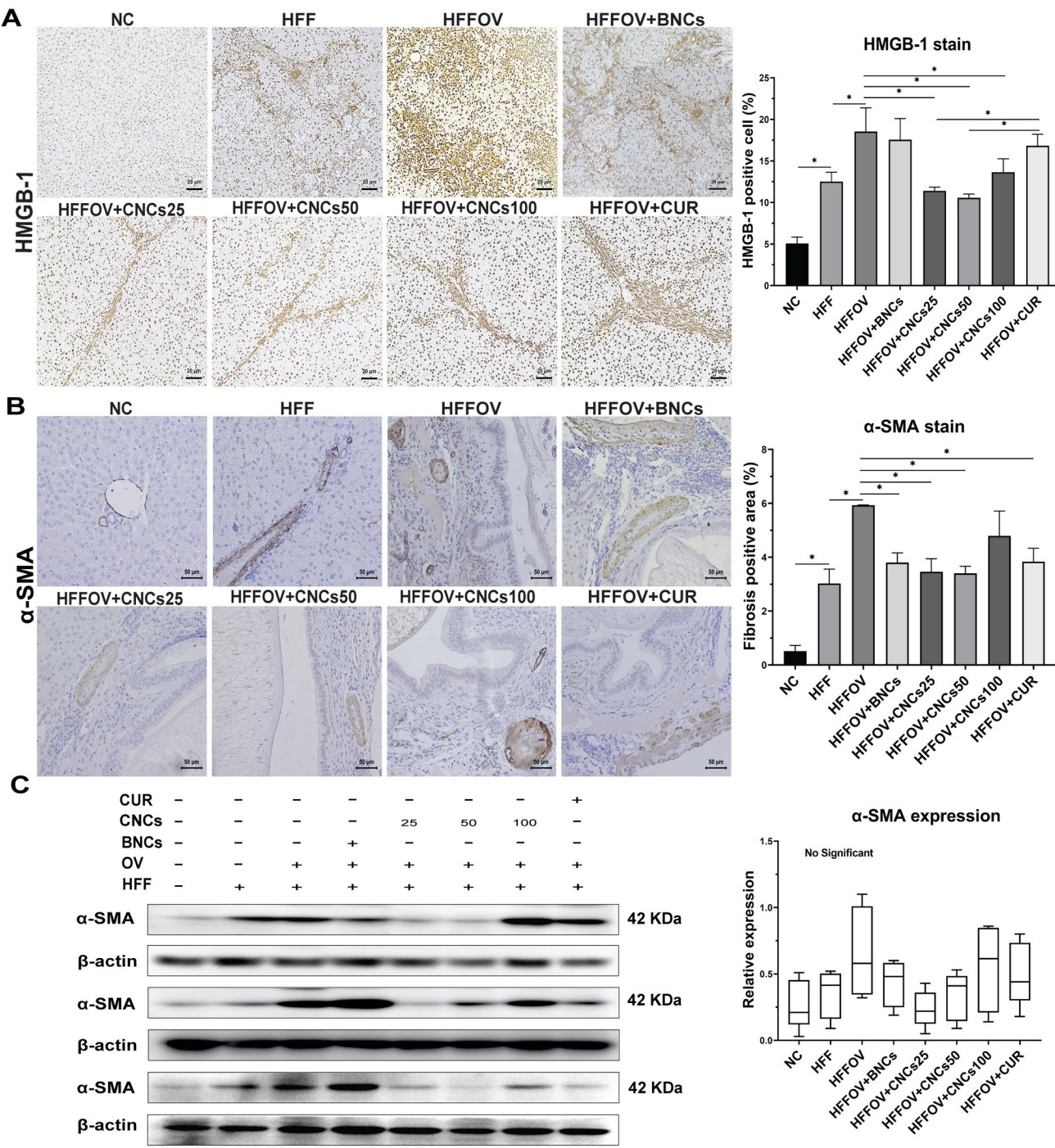

**Fig 6. Effect of treatment with CNCs on inflammation and fibrosis in hamsters infected with *O. viverrini* and fed an HFF diet.** A): Images show immunohistochemical staining (original magnification ×100) for HMGB–1 on the left and a graph of intensity of cell inflammation (assessed at 200x magnification) on the right. B): Images show immunohistochemical staining (original magnification ×200) for α–SMA on the left and a graph of intensity of cell inflammation (assessed at 200x magnification) on the right. C): Protein expression of α-SMA by western blot and quantitative analysis was normalized by β-actin from different individual hamsters. Results are presented as mean±SD. Statistical analyses were done using one-way ANOVA *$P <$ 0.05 (Tukey's multiple comparisons test) (n = 4–5 in each group). Abbreviations are the same as in Fig 3 legend.

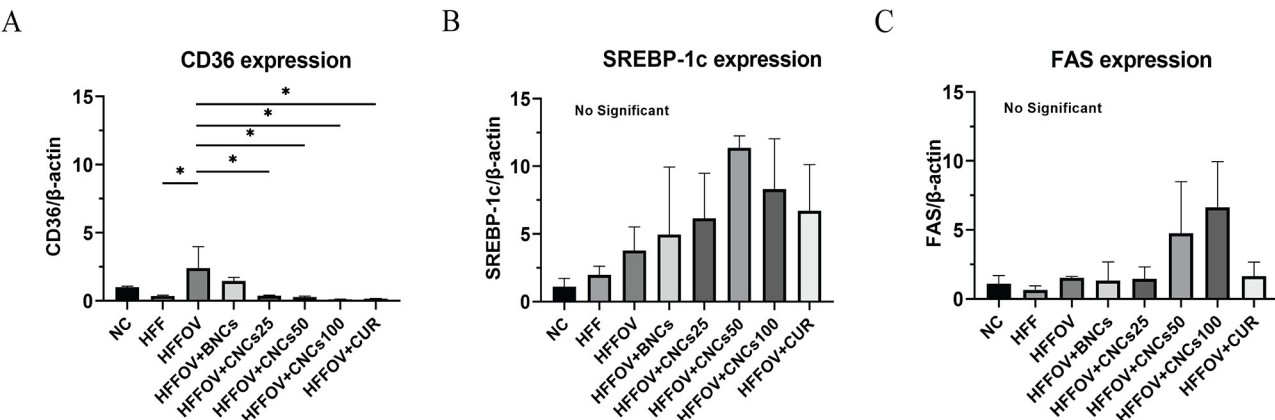

**Fig 7. Effects of curcumin-loading nanocomplexes on mRNA expressions levels.** A): CD36 associated with free fatty-acid uptake by liver cells; B): SREBP-1c; and C): FAS associated with lipid synthesis. Expression levels were measured at the end of the 3-month experimental period. For each group, total RNA was measured from the tissues of three hamsters and the mean±SD shown. Statistical analyses were done using one-way ANOVA *$P < 0.05$ (Tukey's multiple comparisons test). Abbreviations are the same as in Fig 3 legend.

These indicate that the treatments we used might not do much to improve morphological features of the liver.

Treatment with CNCs (all doses) led to decreased liver pathology by reducing fat accumulation, ballooning, inflammation, and fibrosis in both lobular and hilar areas of the liver. The levels of ALT, TC, and TG from blood serum samples were also significantly lower. These findings were similar to those in previous experimental and clinical studies using native curcumin treatment [25, 34, 35]. It has been postulated that curcumin ameliorates hepatic steatosis and reduces elevated biochemical parameters through removal of fat from liver by decreasing synthesis of triglyceride and other lipids, and lowering inflammation [36, 37], as well as improving levels of liver enzymes and insulin resistance [38]. Moreover, reduction of triglyceride storage is related to de-novo lipolysis, the key player in the process of assembling very low-density lipoprotein (VLDL) which is secreted out of the liver [39]. Unexpectedly, hamsters in the treatment group given blank nanocomplexes (BNCs) exhibited reduced levels of ALT and TC, indicating that the nanomaterials alone might be partially involved in attenuation of fat accumulation in liver. We suspect that macrophages might phagocytose BNCs and excrete them into the blood circulation [26]. However, further study is required in this regard.

We evaluated the effects of CNCs on the expression levels of genes related to uptake of fatty acids (CD36 gene) and to the TG synthesis pathway [40]. The expression level of CD36 was decreased after treatment with CNCs or native curcumin. These results were similar to those previously reported by Mun and colleagues after treatment with water-extracted turmeric of mice fed a high-fat diet for eight weeks [41]. Moreover, the reduced expression of CD36 in our study was also associated with the lowering of serum TG levels, implying that CNCs might have an effect on TG synthesis by reducing free fatty-acid uptake.

Curcumin has known antioxidant and anti-inflammatory properties by suppressing oxidative stress, ER stress, cell damage, and collagen deposition, as well as inhibiting liver damage by reducing the cytosolic and nuclear translocation of high mobility groups, especially high mobility group box 1 (HMGB–1) [42]. In this study, we also found a significantly decreased expression of HMGB–1 in the CNCs treatment group, especially at doses of 25 and 50 mg/kg bw, while levels in groups treated with native curcumin and BNCs were not statistically significantly different from the HFFOV group. This finding indicated that CNCs had a higher ability to reduce HMGB–1 expression than did native curcumin. The low efficiency of native

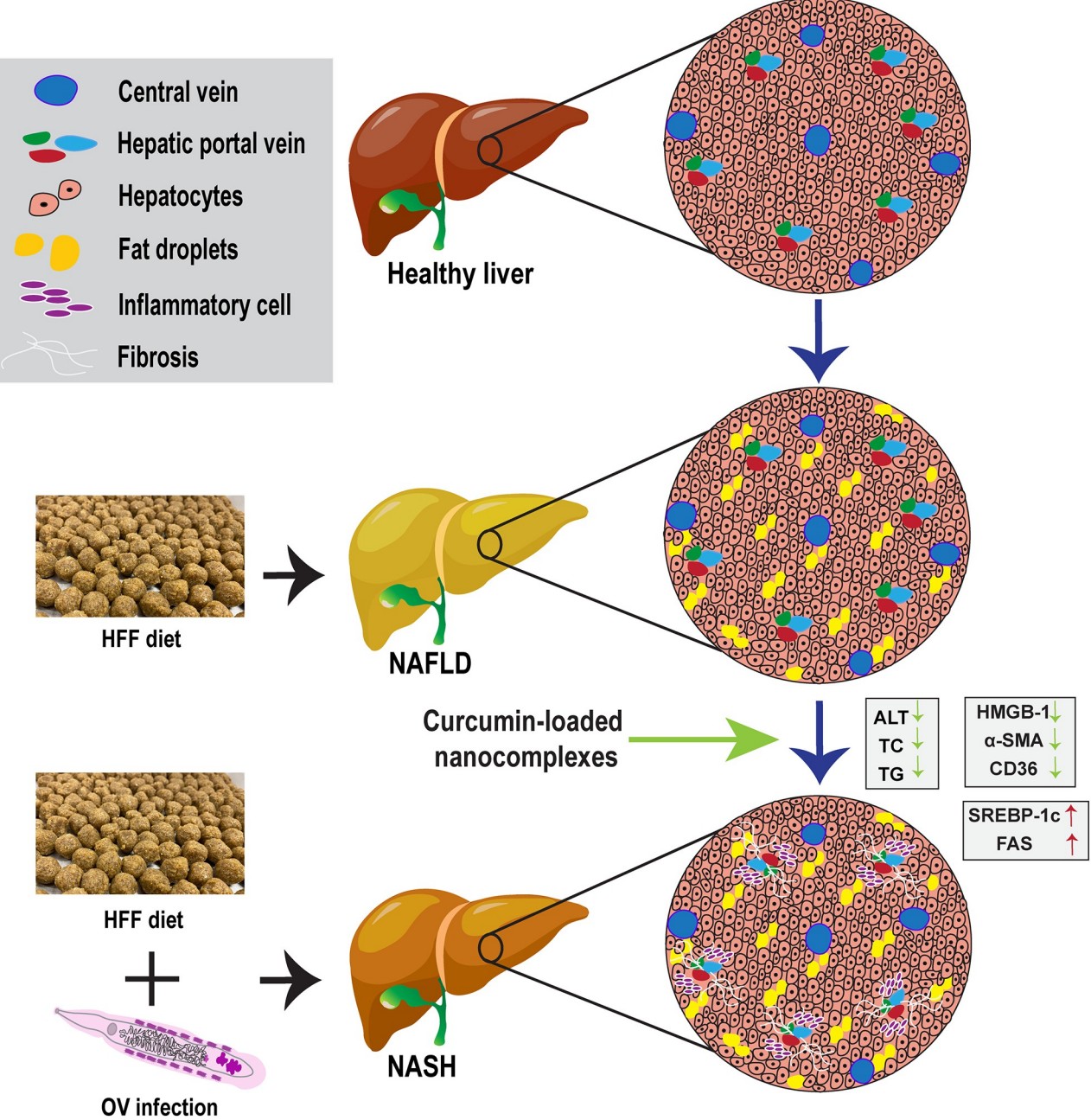

**Fig 8. Potential mechanism of curcumin-loaded nanocomplexes ameliorate the severity of NASH in hamsters infected with *Opisthorchis viverrini*.**

curcumin for reducing inflammation levels might be due to its water insolubility, instability, and poor bioavailability [23]. In addition, the expression level of α–SMA, the marker for fibrosis, was also lower in groups treated with CNCs and native curcumin. These findings are consistent with previous research using native curcumin or nanocurcumin treatments that could lower the levels of HMGB–1 and α–SMA expression [21, 29, 43].

Surprisingly, we found that SREBP–1c and FAS genes were up-regulated in all treatment groups and the highest expression was found in the CNCs treatment groups, followed by native curcumin and BNCs groups in that order. The sterol regulatory element binding

protein-1c (SREBP-1c) regulates expression of several enzymes implicated in cholesterol, lipid and glucose metabolism [44]. Increased SREBP-1c expression is responsible for initiating the transcription of lipogenic enzymes for catalyzing TG production and can induce insulin resistance [39]. Our results contrasted with those of previous studies, which have indicated that SREBP–1c expression was reduced after curcumin treatment [21, 43]. These contrasting findings may be due to differences in experimental designs: we administered CNCs treatments as well as other supplements in our study less frequently than previous researchers who fed high doses of curcumin on a daily basis [43, 45]. Although we found up-regulation of SREBP–1c and FAS genes, our histopathological study strongly indicated lower severity of hepatic disease in the treatment groups. Possibly, the homeostasis of lipid metabolism in the liver might be precisely controlled by several metabolic pathways, such as those associated with lipoprotein metabolism, cholesterol metabolism, bile acid metabolism and fatty acid metabolism [46, 47]. However, the underlying mechanism by which curcumin acts on hepatic lipid metabolism needs further research.

Nevertheless, although CNCs treatment did not show a dose-dependent effect on several results, treatment with CNCs, particularly at 50 mg/kg bw, three times per week, demonstrated a greater ability to reduce hepatic pathology severity and related biochemical parameters than CNCs at 25 and 100 mg/kg bw, or native curcumin or BNCs. This concentration of CNCs (50 mg/kg bw) might be the appropriate dose for use in further studies that are required.

## Conclusion

This study indicates that the severity of hepatic injury and NASH were alleviated by CNCs treatment. Supplementation with CNCs also reduced both plasma triglyceride and cholesterol concentrations. These beneficial effects of CNCs treatment were partly mediated by reducing the expression of genes associated with fatty-acid uptake (CD36), inflammation (HMGB–1) and fibrogenesis (α–SMA). This study demonstrates the therapeutic potential of CNCs that might have future clinical application for averting NASH development.

## Supporting information

**S1 Table. Nutrient composition of HFF diet.**
(DOCX)

**S1 Fig. Raw gel images for α-SMA and β-actin proteins expression.**
(PDF)

## Acknowledgments

We would like to acknowledge Prof. David Blair for editing a manuscript via publication clinic, KKU.

## Author Contributions

**Conceptualization:** Lakhanawan Charoensuk, Kitti Intuyod, Somchai Pinlaor.

**Data curation:** Chutima Sitthirach, Lakhanawan Charoensuk, Chawalit Pairojkul, Apisit Chaidee, Thatsanapong Pongking, Phonpilas Thongpon, Sakda Waraasawapati.

**Formal analysis:** Chutima Sitthirach, Lakhanawan Charoensuk, Chawalit Pairojkul, Apisit Chaidee, Nuttanan Hongsrichan, Sakda Waraasawapati, Manachai Yingklang.

**Funding acquisition:** Somchai Pinlaor.

**Methodology:** Chutima Sitthirach, Lakhanawan Charoensuk, Apisit Chaidee, Thatsanapong Pongking, Phonpilas Thongpon, Chanakan Jantawong, Somchai Pinlaor.

**Project administration:** Lakhanawan Charoensuk, Somchai Pinlaor.

**Supervision:** Lakhanawan Charoensuk, Kitti Intuyod, Nuttanan Hongsrichan, Somchai Pinlaor.

**Validation:** Chutima Sitthirach, Lakhanawan Charoensuk, Chawalit Pairojkul, Apisit Chaidee, Kitti Intuyod, Thatsanapong Pongking, Phonpilas Thongpon, Chanakan Jantawong, Nuttanan Hongsrichan, Sakda Waraasawapati, Manachai Yingklang, Somchai Pinlaor.

**Visualization:** Chutima Sitthirach, Lakhanawan Charoensuk, Chawalit Pairojkul, Apisit Chaidee, Kitti Intuyod, Thatsanapong Pongking, Phonpilas Thongpon, Chanakan Jantawong, Sakda Waraasawapati, Somchai Pinlaor.

**Writing – original draft:** Chutima Sitthirach, Lakhanawan Charoensuk, Manachai Yingklang.

**Writing – review & editing:** Chutima Sitthirach, Lakhanawan Charoensuk, Chawalit Pairojkul, Apisit Chaidee, Kitti Intuyod, Thatsanapong Pongking, Phonpilas Thongpon, Chanakan Jantawong, Nuttanan Hongsrichan, Sakda Waraasawapati, Manachai Yingklang, Somchai Pinlaor.

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
