## [Decision Letter · Decision Letter 0]

18 Jul 2022

PONE-D-22-11833Curcumin-loaded nanocomplexes ameliorate the progression of NAFLD to NASH in hamsters infected with Opisthorchis viverriniPLOS ONE

Dear Dr. Pinlaor,

Thank you for submitting your manuscript to PLOS ONE. After careful consideration, we feel that it has merit but does not fully meet PLOS ONE’s publication criteria as it currently stands. Therefore, we invite you to submit a revised version of the manuscript that addresses the points raised during the review process. We have received a thoughtful review of your manuscript. In order to not unduly lengthen the review process, I would like to give you the opportunity to revise your manuscript based on the one review. I agree with the points raised by the reviewer.

We look forward to receiving your revised manuscript.

Kind regards,

Michael W. Greene, Ph.D.

Academic Editor

PLOS ONE

Journal Requirements:

“This study was supported by the Research and Researchers for Industries (RRI: MSD62I0041), Thailand Science Research and Innovation (TSRI: RDG6250045) and Mekong Health Science Research Institute Khon Kaen University (MeHSRI09/2561).”

Reviewers' comments:

Reviewer's Responses to Questions

**Comments to the Author**

1. Is the manuscript technically sound, and do the data support the conclusions?

Reviewer #1: Partly

2. Has the statistical analysis been performed appropriately and rigorously? 

Reviewer #1: Yes

3. Have the authors made all data underlying the findings in their manuscript fully available?

Reviewer #1: Yes

4. Is the manuscript presented in an intelligible fashion and written in standard English?

Reviewer #1: Yes

5. Review Comments to the Author

Reviewer #1: The authors investigated the effect of nanoencapsulated and natural curcumin on liver injuries and the development of nonalcoholic fatty-liver disease in combination with OV liver fluke infection and a high-fat high-fructose diet. The strength of the study is the number of groups, including three different doses of nanoencapsulated curcumin and the number of animals within the groups (7 individuals). In general, the authors show the results of the original research and the data obtained may be of interest to the researchers in similar areas. Nevertheless, I believe that the text of the manuscript needs a revision.

1. Fig.5 - there is not any indications on diagrams whether something is changing significantly or not. Accordingly, only those results that are significant should be described in the text, so (lines 272-281) ...” The expression level of CD36 was high in the HFFOV group compared with the HFF group (Fig 5A) and was lower than this in all groups receiving treatment (with BNCs, CNCs or CUR).” Are there statistically significant differences between each of these groups to draw such conclusions?

2. Similar question applies to Fig4C - is there any significant difference among the groups? If there are no statistical changes, then there should also be no reason to indicate in the text that one group is higher or lower. At present, in the results, discussion, and even in the abstract, some text states about non-significant changes between groups. Authors should carefully revise the text of the manuscript and remove all such statements about statistically insignificant changes. Moreover, where there is no significant difference, it could be written that there were no significant differences among groups. Otherwise, it will be a misinterpretation of the obtained results.

3. Table 2 can be transferred to the supplements

4. It is unclear, what criteria the authors used to clearly distinguish NAFLD and NASH, could the author provide any criteria in Table 3 how it is possible to separate these two pathologies using the hamster model? It would be very helpful if the authors modified the Table 3 and inserted NAFLD and NASH criteria in the histological scoring system

5. There are many abbreviations in the abstract, while some of them are given without decoding, for example, HFF diet (HFF); TC and TG.

6. The company name and catalogue code of the primary and secondary antibodies used should be indicated in the M&M.

6. PLOS authors have the option to publish the peer review history of their article (what does this mean?). If published, this will include your full peer review and any attached files.

Reviewer #1: No

---

## [Author Response · Author response to Decision Letter 0]

18 Aug 2022

PONE-D-22-11833

Curcumin-loaded nanocomplexes ameliorate the progression of NAFLD to NASH in hamsters infected with Opisthorchis viverrini

Title has been changed to “Curcumin-loaded nanocomplexes ameliorate the severity of nonalcoholic steatohepatitis in hamsters infected with Opisthorchis viverrini”

Response to the Journal Requirements:

Response: We have rechecked and ensured that the manuscript meets PLOS ONE’s style requirements, including those for file naming. 

Response: Our Data Availability statement can now be found as supportive information. 

“This study was supported by the Research and Researchers for Industries (RRI: MSD62I0041), Thailand Science Research and Innovation (TSRI: RDG6250045) and Mekong Health Science Research Institute Khon Kaen University (MeHSRI09/2561).”

Response: we have stated the role of funders as suggested: “The funders had no role in study design, data collection and analysis, decision to publish, or preparation of the manuscript.”

Response: We have provided our raw blot/gel image data in Supporting information as is required.

Response: we have re-checked all references and all are correct.

Response to the Reviewers' comments:

Reviewer #1: The authors investigated the effect of nanoencapsulated and natural curcumin on liver injuries and the development of nonalcoholic fatty-liver disease in combination with OV liver fluke infection and a high-fat high-fructose diet. The strength of the study is the number of groups, including three different doses of nanoencapsulated curcumin and the number of animals within the groups (7 individuals). In general, the authors show the results of the original research and the data obtained may be of interest to the researchers in similar areas. Nevertheless, I believe that the text of the manuscript needs a revision.

Response: We thank the reviewer for their critical evaluation and suggestions for improving the manuscript. We would like to respond to each point as follows.

1. Fig.5 - there is not any indications on diagrams whether something is changing significantly or not. Accordingly, only those results that are significant should be described in the text, so (lines 272-281) ...” The expression level of CD36 was high in the HFFOV group compared with the HFF group (Fig 5A) and was lower than this in all groups receiving treatment (with BNCs, CNCs or CUR).” Are there statistically significant differences between each of these groups to draw such conclusions?

Response: We have reanalyzed and revised Fig 5A (old version, now Fig 7A in the new version). Text was also revised in line with this change. 

2. Similar question applies to Fig4C - is there any significant difference among the groups? If there are no statistical changes, then there should also be no reason to indicate in the text that one group is higher or lower. At present, in the results, discussion, and even in the abstract, some text states about non-significant changes between groups. Authors should carefully revise the text of the manuscript and remove all such statements about statistically insignificant changes. Moreover, where there is no significant difference, it could be written that there were no significant differences among groups. Otherwise, it will be a misinterpretation of the obtained results.

Response: We have reanalyzed and revised Fig 4 (old version, now Fig 6 in the new version). Text was also revised in line with this change.

3. Table 2 can be transferred to the supplements

Response: We have moved Table 2 to the supplements where it is referred to as Table S1.

4. It is unclear, what criteria the authors used to clearly distinguish NAFLD and NASH, could the author provide any criteria in Table 3 how it is possible to separate these two pathologies using the hamster model? It would be very helpful if the authors modified the Table 3 and inserted NAFLD and NASH criteria in the histological scoring system.

Response: We have revised Table 3 (old version, now Table 2 in the new version) and inserted the criteria used to distinguish between NAFLD and NASH as suggested.

5. There are many abbreviations in the abstract, while some of them are given without decoding, for example, HFF diet (HFF); TC and TG.

Response: We have amended the abstract accordingly in the revised version. 

6. The company name and catalogue code of the primary and secondary antibodies used should be indicated in the M&M.

Response: We have added this information in the revised version.

---

## [Decision Letter · Decision Letter 1]

13 Sep 2022

Curcumin-loaded nanocomplexes ameliorate the severity of nonalcoholic steatohepatitis in hamsters infected with *Opisthorchis viverrini*

PONE-D-22-11833R1

Dear Dr. Pinloar,

We’re pleased to inform you that your manuscript has been judged scientifically suitable for publication and will be formally accepted for publication once it meets all outstanding technical requirements.

Kind regards,

Michael W. Greene, Ph.D.

Academic Editor

PLOS ONE

Additional Editor Comments (optional):

Reviewers' comments:

Reviewer's Responses to Questions

**Comments to the Author**

1. If the authors have adequately addressed your comments raised in a previous round of review and you feel that this manuscript is now acceptable for publication, you may indicate that here to bypass the “Comments to the Author” section, enter your conflict of interest statement in the “Confidential to Editor” section, and submit your "Accept" recommendation.

Reviewer #1: All comments have been addressed

2. Is the manuscript technically sound, and do the data support the conclusions?

Reviewer #1: Yes

3. Has the statistical analysis been performed appropriately and rigorously? 

Reviewer #1: Yes

4. Have the authors made all data underlying the findings in their manuscript fully available?

Reviewer #1: Yes

5. Is the manuscript presented in an intelligible fashion and written in standard English?

Reviewer #1: Yes

6. Review Comments to the Author

Reviewer #1: The authors took into account my suggestions and comments and made appropriate changes to the manuscript. The text has been significantly improved and technically sound. I have no further suggestions.

7. PLOS authors have the option to publish the peer review history of their article (what does this mean?). If published, this will include your full peer review and any attached files.

Reviewer #1: No

---

## [Editor Report · Acceptance letter]

19 Sep 2022

PONE-D-22-11833R1 

Curcumin-loaded nanocomplexes ameliorate the severity of nonalcoholic steatohepatitis in hamsters infected with *Opisthorchis viverrini*

Dear Dr. Pinlaor:

I'm pleased to inform you that your manuscript has been deemed suitable for publication in PLOS ONE. Congratulations! Your manuscript is now with our production department. 

Kind regards, 

on behalf of

Dr. Michael W. Greene 

Academic Editor

PLOS ONE